# Morphology of the Ventral Process of the Sixth Cervical Vertebra in Extinct and Extant *Equus*: Functional Implications

**DOI:** 10.3390/ani13101672

**Published:** 2023-05-17

**Authors:** Sharon May-Davis, Robert Hunter, Richard White

**Affiliations:** 1Canine and Equine Research Group, University of New England, Armidale, NSW 2351, Australia; rjhunter120@gmail.com; 2Mammoth Site, Hot Springs, SD 57747, USA; rswhite@mammothsite.org

**Keywords:** caudal ventral tubercle (CVT), cranial ventral tubercle (CrVT), *Equus ferus caballus*, *longus colli*, sixth cervical vertebra (C6), ventral tubercle

## Abstract

**Simple Summary:**

The mammalian body plan is like a ‘blueprint’ that supports the survival and procreation of a species. This plan includes specialized bony structures for muscle attachment such as the ventral process of C6, a two-part bony projection divided into cranial and caudal ventral tubercles (CVT). In modern horses, muscles such as the *longus colli* attach to the ventral process of C6 and aid in the flexion, stabilization, fixation, and force redirection of cervical vertebrae. Recent studies identified an anomalous variation where the CVT was either unilaterally or bilaterally absent. Therefore, the purpose of this study is to determine if anomalous variations of the CVTs, as described in modern horses, is a post-domestication congenital abnormality or a normal presentation throughout evolution. For this purpose, the ventral processes of C6 in extinct and extant *Equus* specimens in museums and research/educational facilities were examined. The findings revealed the CVT of C6 was present in those extinct and extant specimens examined, and that its absence was only evident in modern horses. This implies that the absent CVT manifested post-domestication. Although the clinical significance is unknown, the relevance might be considered when reporting on diagnostic images of C6, and especially, for potential buyers in pre-purchase cases.

**Abstract:**

In this study, we examined the ventral process of C6 in extinct and extant *Equus* (sister taxa to *Equus ferus caballus* only) with the purpose of describing normal morphology and identifying anomalous variations relevant to recent studies describing a congenital malformation in *E. ferus caballus*. Overall, 83 specimens from 9 museums and 3 research/educational facilities were examined, totalling 71 extinct specimens from 12 species and 12 extant specimens from 5 species. The lateral view revealed that a large convexity exists in the ventral process between the cranial ventral tubercle (CrVT) and the caudal ventral tubercle (CVT) in the earliest ancestor, *Hyracotherium grangeri*, from 55 mya, which receded throughout the millennia to become a smaller convexity in *E. ferus caballus* and the sister taxa. The CrVT is visibly shorter and narrower than the CVT, with a constricted section directly ventral to the transverse process, essentially demarcating the CrVT and CVT. No congenital malformations were evident. As the ventral process of C6 is an integral component for muscle attachment in supporting the head/neck during posture and locomotion, this would indicate that the caudal module in the cervical column might be compromised when a partial or complete absence of the CVT is detected via radiographs in modern *E. ferus caballus*.

## 1. Introduction

In the mammalian neck, there are a wide variety of morphological specialisations due to evolutionary processes, which include three complex cervical modules [1,2,3]. With a few exceptions, the three modules comprise seven cervical vertebrae (CV): cranial (C1 and C2), mid (C3–C5) and caudal (C6 and C7). Each vertebrae in the upper and lower modules present highly specialised conserved traits specifically designed for a diverse range of daily functions [1,2,3]. When conserved traits are shared by a group of phylogenetically related animals, they are referred to as a ‘body plan’ [4]; as demonstrated by the specialised ventral process of C6 in mammals [1,2,3,5,6,7,8,9,10,11,12].

In *Equus ferus caballus*, veterinary texts define C6 as an ‘atypical’ vertebra inclusive of C1, C2 and C7; whereas C3–C5 are ‘typical’ [7,8,12,13,14,15]. The differentiation between CV that are typical and atypical are: a typical CV is relatively unspecialised; while an atypical CV presents highly specialised structural features for soft tissue attachment/s critical for specific head and neck biomechanics [1,2,3,12,13,14,15,16,17,18,19]. In C6, its atypical feature is a pleurapophysis vestigial rib forming a bilateral ventral projection traversing the entire length of the vertebral body [1], which is referred to in the literature by several names; the ventral process, the ventral lamina (*lamina ventralis*), the ventral transverse process and the ventral tubercle [6,7,8,12,13,14,15,16,18,19,20,21,22,23,24,25,26,27,28,29]. For the purpose of this study, the ventral projection will be referred to as the ventral process of C6.

The ventral process of C6 demonstrates two specific osteological features; a cranial ventral tubercle (CrVT) and a caudal ventral tubercle (CVT), and each provides attachments for the deep perivertebral muscle, *longus colli* [7,12,13,14,15,16]. Rombach, Stubbs and Clayton [16] demonstrated that the CrVT in *E. ferus caballus* is the attachment site for the cervical portion of the multi-bundled *longus colli* muscle, while the CVT is the insertion point for the single-bundled thoracal tendon of the *longus colli* muscle that extends caudally to either T5 or T6. This muscle aids in the fixation, stabilisation, rotation and flexion of the CV, and it also acts as a site of force redirection during muscle contraction of either the cranial or caudal vertebrae to C6 [12,13,14,15,18].

In domestic mammals, the literature primarily describes the ventral process of C6 relative to educational instruction or veterinary management [6,7,8,12,20,21,22,23,24,25,26,27,28,29]. Although, an extensive amount of literature on pathologies and peer-reviewed journals describing anomalous presentations are well documented in some domestic mammals [15,19,24,27,28,30,31,32,33,34,35,36,37,38,39,40]; historically, congenital malformations in the fossil record are rare [41]. Therefore, when two domestic species presenting similar congenital malformations in the cervicothoracic junction involving the ventral process of C6 are reported, namely *Bos taurus domesticus* [30,31,32] and *E. ferus caballus* [34,35,36,37,38,39,40], further investigations into the fossil record are warranted.

Clinically, the condition in *B. taurus domesticus* known as complex vertebral malformation (CVM) is fatal, with the animal being either stillborn or spontaneously aborted during gestation [31,32]. Hence, the preservation in the fossil record is unlikely due to the fragmentation of associative epiphyseal growth plates relative to the ventral process of C6 [7], plus poor preservation of the specimen and/or predation [41]. However, in modern *E. ferus caballus*, the condition is not known to be fatal, and the absent CVT on the ventral process of C6 is evident in mature adults, and therefore, observations are likely [34,35,36,37,38,39,40,41]. Although the absent CVT on C6 is yet to be clinically defined, several authors have postulated or documented symptoms conducive to the congenital malformation in case studies [34,38,40]. Therefore, the aim of this study is to ascertain whether the congenital malformation of the absent CVT in C6 is a recent event in modern *E. ferus caballus* or a normal variation in the population by ascertaining its presence or absence in pre-domesticated *Equus*.

To examine these questions further, this study will examine the fossil record of extinct and extant Equidae. It would be expected that this evidence, if evident, would be determinable as other anomalous variations of the CVT in C6 have been reported in the fossil record. For example, researchers have noted cervical rib facets on the CVT of C6 in Pleistocene *Coelodonta antiquitatis* [42] and *Mammuthus primigenius* [43] and a complete absence of the right CVT in C6 of *Dendrohyrax arboreus* [44]. Therefore, the purpose of this study is to determine if anomalous variations of the ventral process in C6 associated with extant modern *E. ferus caballus* is strictly a post-domestication abnormality with possible functional implications, or if it is a normal variation in the population.

## 2. Materials and Methods

### 2.1. Ethical Statement

No equids were euthanized for the purpose of this study, and observational research was conducted from specimens in museum collections and educational facilities.

### 2.2. Terminology

Nomenclature was derived from Getty [7], Popesko [20] and Gasse [45]. When a specific term for a structural profile or shape required further descriptive terminology, this was derived from either biological or botanical nomenclature or previous literature.

### 2.3. Materials

Twenty-five applications to examine extinct and extant *Equus* collections were made to various museums and research/educational facilities. Collections only housing modern *E. ferus caballus* were excluded, and for logistical reasons related to the availability and suitability of the collection, only nine museums and three educational/research facilities were utilised for this study. Museums—American Museum of Natural History, New York, NY, USA (AMNH); Monrepos Museum for Human Behavioural Evolution, Neuwied, Germany (MMHBE); Natural History Museum, Berlin, Germany (NHMB); Natural History Museum of Los Angeles County, Los Angeles, CA, USA (LACM); Oxford University Museum of Natural History, Oxford, UK (OUMNH); Naturalis Biodiversity Center, Leiden, The Netherlands, (NBC); United States National Museum, Washington, DC, USA (USNM); The Page Museum Rancho La Brea, Los Angeles, CA, USA (RLB); Yokohama Horse Museum, Yokohama, Japan (YHM). Educational facilities included the Australian College of Equine Podiotherapy, Yarck, Australia (ACEPT); Equine Studies, Asch, The Netherlands (ES); the University of Florida, Gainesville, FL, USA (UF).

Peer-reviewed publications that nominated catalogued specimens presenting evidence of complete ventral processes of C6 were also examined.

To be eligible for the study, the specimens required to have suffered from a minimal amount of damage to at least one ventral process where a clear structural definition of the CrVT and CVT was determined. The purpose was to ascertain a normal and or anomalous presentation (Figure 1).

Among those specimens where the ventral process is clearly divided into a CrVT and CVT, such as *Pliohippus pernix* (Figure 2), and where portions of either the CrVT or CVT on one side have been lost to various pressures, such as taphonomy or predation, a revised observation was documented outlining the intact tubercles.

Collectively, the museums, educational/research facilities and peer-reviewed literature yielded 68 extinct *Equus* specimens, 12 extant *Equus* specimens (not *E. ferus caballus*) and 3 peer reviewed publications describing specimens, respectively, totalling 83 specimens. The age of each specimen at expiration was defined by the complete ossification of the caudal epiphyseal growth plate on C6 and subsequently included. Adults were defined by a closed epiphyseal growth plate with no delineation, while subadults were determined by incomplete ossification of the epiphyseal growth plate and obvious delineation. When the classification of a species was undifferentiated, the specimen was labelled as *Equus sp*.

In museum specimens, 12 species of extinct *Equus* were examined, totalling 68 specimens with geologic dates. Eighteen specimens were undifferentiated and categorised as *Equus sp*. according to the museum’s records (Table 1).

In extant *Equus*, five non-*E. ferus caballus* species were examined with one undifferentiated specimen, totalling 12 specimens. In museums, specimens were recorded according to their catalogue number in the collection, except for one (NBC—*Equus sp*.; mounted skeleton that was not *E. ferus caballus*). In educational facilities, the specimens were classified by the animal’s name pre-mortem (Table 2).

Gidley (1903), Wood et al. (2011) and Franzen and Haberstzer (2017) presented three peer-reviewed publications describing three extinct *Equus* species, *Neohipparion whitneyi, Hyracotherium grangeri* and *Eurohippus messelensis*, respectively (Table 3) [46,47,48].

### 2.4. Methods

The ventral process of C6 was observed and described, and the left or right lateral profile/s were digitally recorded in extinct and extant species of *Equus* (not *E. ferus caballus*).

The assigned shape/s describing the lateral profile of the ventral process of C6 was/were derived using either botanical and/or biological nomenclature or from previous publications describing CV.

## 3. Results

### 3.1. General Anatomy

In all 83 specimens, the ventral process of C6 appeared on the lateral ventral border in an antero-posterior orientation, separate and distal to the transverse process. In 82/83, the CrVT was anterior to the transverse process, while the CVT remained posterior. *Eurohippus messelensis* was the exception and did not present a CrVT nor CVT according to the depictions and radiographs [48]. Excluding *Eurohippus messelensis*, in 82 specimens, the lateral profile of the ventral process depicts a separation between the CrVT and CVT that is noted by a convexity in the morphology directly distal to the transverse process. The extent of convexity was not uniform between specimens and ranged from lesser convexity between the CrVT and CVT to distinct hourglass convexity (Figure 1a and Figure 2).

From the ventral view, 82 specimens presented a tube-like morphology of the ventral process, with a central constriction distal to the transverse process. Enthesis patterns were evident on both tubercles in relation to the attachment of the *longus colli* muscle, i.e., the cranial *longus colli* attachment to the CrVT and the thoracal tendon of the *longus colli* to the CVT. The CVT appeared to be more expanded than the CrVT, and when both cranial and caudal ventral tubercles were present, the overall outside width from the left to right tubercles was greater across the CrVTs (Figure 3).

No congenital malformations or anomalous presentations such as those previously described were evident in the ventral process of C6 in the 83 specimens, nor in those specimens that were deemed to be unsuitable for the study.

### 3.2. Lateral Profile of the Ventral Process

In all 82/83 specimens, the CVT appeared to be longer than the CrVT. The size of convexity in the ventral process presented three distinct morphological variations that were determined by size, and simply described as being small, medium or large (Figure 4).

In extinct *Equus*, 71 specimens from museums and the literature presented a total of 38 small, 19 medium, and 13 large convexities, with *Eurohippus messelensis* presenting with a morphologically different bilateral ventral process with no convexities (Table 4).

*Eurohippus messelensis* presented with bilateral ventral processes with no convexities, which appeared to be well-developed ventrolateral appendages from the vertebral body [48], similar to a trapezoid in profile view. Gidley’s (1903) description of the ventral process describes the ventral surface of the 6th cervical as flat, turning laterally downward into the wing-like transverse processes, which are more strongly developed than they are in *Equus* [46]. Previous literature describes these as cranial and caudal ventral transverse processes [6].

All 12 specimens in extant *Equus* presented small convexities (Figure 1a) in the ventral process; yet, the demarcation between the CrVT and CVT was still clear (Table 5).

### 3.3. Variations of the Eqpiphyseal Growth Plate

During the examination of extinct and extant *Equus* variations in the CVT, the epiphyseal growth plate could be noted in subadults as being evident or absent. The absence was not anomalous variation, but a portion of the epiphyseal growth plate that had been lost due to destructive process, such as degradation or human intervention during the preparation of the vertebra (Figure 5).

In the extant specimen of Przewalskii’s horse from Equine Studies (Rideg), the caudal epiphyseal growth plate was intact upon dissection and maceration; its loss can be attributed to final preparations, such as degreasing.

## 4. Discussion

In this study, we examined the morphology of the ventral process of C6 in extinct and extant *Equus.* The findings revealed that the mammalian body plan for this specialised atypical structure remained a conserved trait in Equidae from its earliest ancestor, *Hyracotherium grangeri* [47]. The only noted variation in morphology was the size of the convexity between the CrVT and CVT, receding from large to small over 55 million years, except in *Eurohippus messelensis* (47 mya), an extinct European descendant from the first North American migration [48].

The relationship between the convexity of the ventral process and its decrease in size over the millennia might be functionally related to the increase in size of the head/neck, change in feeding regimes from predominantly browsing to predominantly grazing and/or environmental pressures, such as predation [49]. The altered morphology of the ventral process led to an elongated CrVT and CVT with a reduced convexity, thus providing a longer attachment site for the *longus colli* muscle to adequately support head/neck functions. However, to verify this, future morphometric studies must be conducted.

As for the variations seen in individual cervical vertebrae (C1–C7), they can be described in terms of intravertebral versus intervertebral modules and defined as a component of a developing organism (e.g., an embryo) that is semi-autonomous relative to pattern formation and differentiation [50,51]. Arnold (2021) mapped out the different possible modular schemes for cervical vertebrae based on developmental, morpho-functional and paleontological perspectives [1]. Similarly, Randau and Goswami (2017) demonstrated that patterns of shape integration reflect modular organisation in felids [51], while in other carnivores, larger modules support integration between adjacent vertebrae to meet locomotory and functional demands [52].

In Equidae, as per most mammals, the cervical spine is held as vertically as possible to reduce the distance between the weight of the head and the sustaining cervicothoracic junction (C5–T2) [1,2]. This leads to the stereotypical vertical, s-shaped and self-stabilising resting posture of the cervical spine [53,54]. When head/neck movements are initiated from this posture, orientation and gaze changes in the sagittal plane are restricted to the occiput between C2 module and C6–T1 module [1,2]. Arnold (2021) stated that this functional modularity is supported by two prominent bony processes in mammals that provide major muscle attachment sites for head and neck motions: the enlarged spinous process on C2, and the ventral process of C6 [1]. These findings support Bainbridge’s (2018) comments that the ventral process of C6 acts as a site of force redirection during muscle contraction of the cranial and/or caudal vertebrae to C6 [15].

In modern *E. ferus caballus*, the ventral process of C6 being unilaterally or bilaterally absent has been reported in multiple breeds from numerous countries, for example, Australian Thoroughbreds (38%) and Dutch Warmbloods (33%) [15,19,33,34,35,36,37,38,39,40]. Clinical symptoms are yet to be defined in longitudinal studies; yet, this presentation across breeds and geographical regions suggests a familial connection [34]. With respect to the concepts of cervical modularity, this anomalous variation in the ventral process of C6, which is not seen in ancestral *Equus*, infers that head and neck functions could be compromised through the limitation or lack of the attachment site of the thoracal *longus colli* muscle. Through case studies based on the gross examination of the thoracal *longus colli* muscle, May-Davis and Walker (2015) concurred that it either relocated with an altered tendon morphology or hypertrophied on the affected side. The authors also described potential symptomatic observations of afflicted horses bearing this anomalous variation [40]. The authors of similar studies concurred with these findings and further described ataxic behaviours [34,38,55].

The findings from this study indicate that the anomalous variations reported in the ventral process of C6 in modern *E. ferus caballus* is a recent event and is not indicative of normal variation within the population of *Equus*, nor was it present in pre-domestic *Equus* or ancestral *Equus.* However, the limitations in this study were identified as the small number of extant *Equus* specimens (non *E. ferus caballus*) and the lack of complete specimens in the fossil record and access to further specimens. Even so, the evidence to date suggests that this anomalous variation in the ventral process in C6 is not present in the fossil record (*n* = 71). Therefore, in modern *E. ferus caballus*, when the CVT of C6 is absent, functional ramifications could be a concern when the structural integrity of the caudal cervical module in head/neck function is compromised. These findings might benefit equine practitioners reporting on diagnostic images of C6, and especially, for potential buyers in pre-purchase cases.

## 5. Conclusions

The ventral process of C6 is a conserved trait in most mammals, and its highly specialised atypical structure is present in extinct and extant *Equus.* Functionally, the muscles attached to this process aid in the flexion, stabilization, fixation and force redirection of the cervical vertebrae, which are essential for head/neck posture and locomotion. When the CVT of the ventral process is absent, as reported in modern *E. ferus caballus*, there could be consequences for normal functions. This study provides evidence that absent CVTs are not present in extinct *Equus*, nor among a limited number of extant species of *Equus*, and they are only present in *E. ferus caballus.*

Therefore, as an integral part of the caudal cervical module, any anomalous variations in the ventral process of C6 might lead to dysfunctional ramifications of the cervicothoracic junction. Future morphological and biomechanical studies would need to be conducted to understand the full implications.

## Figures and Tables

**Figure 1 animals-13-01672-f001:**
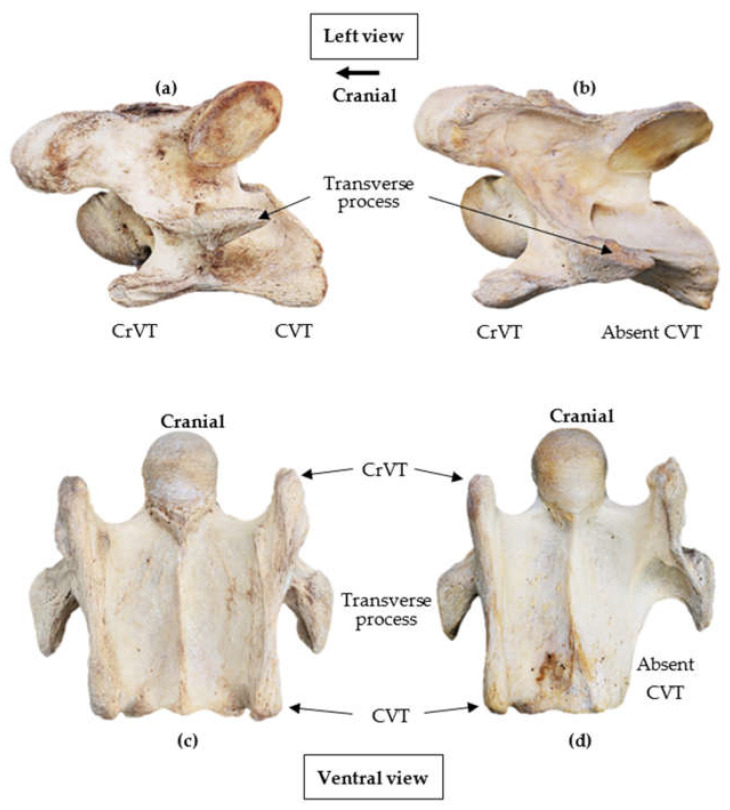
Normal and anomalous presentations of the 6th cervical vertebra in modern *Equus ferus caballus*. (**a**,**c**) normal presentation; (**b**,**d**) anomalous presentation of an absent left caudal ventral tubercle (Not to scale) (Photographs courtesy of Equus Soma, Aiken SC).

**Figure 2 animals-13-01672-f002:**
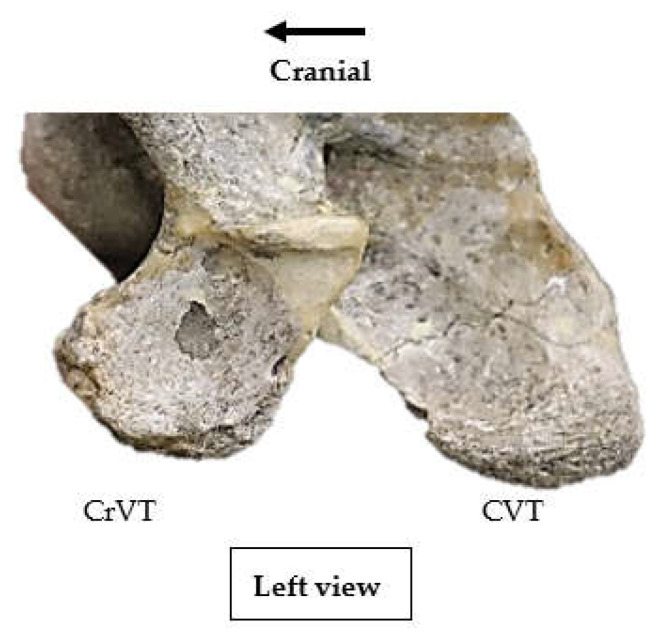
The left ventral process of C6 in *Pliohippus pernix*, AMNH catalogue no. 60810, presenting a clearly divided cranial and caudal ventral tubercle (Not to scale).

**Figure 3 animals-13-01672-f003:**
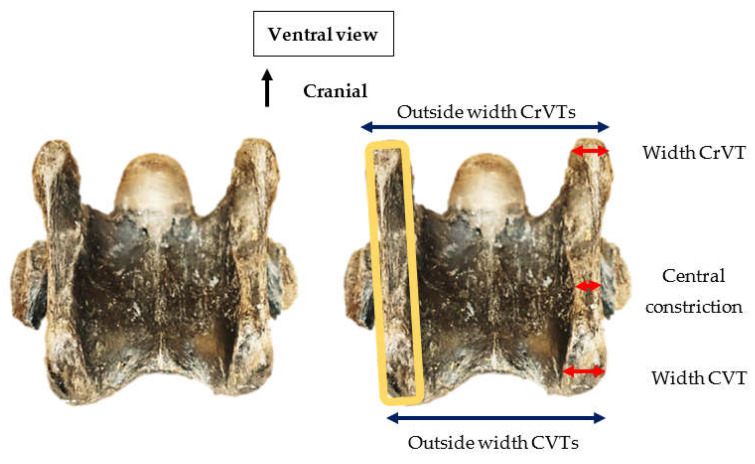
Ventral view of the 6th cervical vertebra in *Equus sp*. Leisey 1A B-Y UF 82283 depicting the tube-like formation of the tubercles (orange lineation), widths of cranial ventral tubercle, caudal ventral tubercle, and central constriction (red arrows), and the overall outside width between the left and right cranial ventral tubercules and caudal ventral tubercles (black arrows) (Not to scale).

**Figure 4 animals-13-01672-f004:**
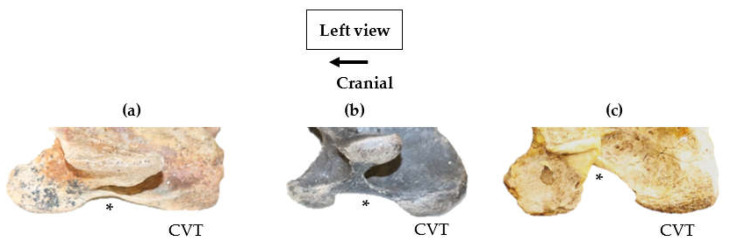
Left lateral profile of the ventral process of the 6th cervical vertebra in extinct *Equus;* the convexity of morphology between the cranial ventral tubercle and the caudal ventral tubercle is noted (*****). (**a**) Small-sized convexity—*Equus simplicidens* USNM PAL 785556. (**b**) Medium-sized convexity—Leisey 1A UF 151092. (**c**) Large-sized convexity—*Pliohippus pernix* 60803 (Not to scale).

**Figure 5 animals-13-01672-f005:**
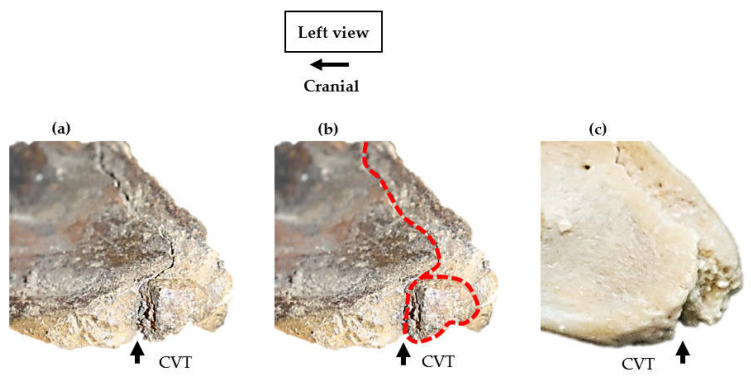
Degradation and partial loss (black arrow) of the left epiphyseal in the caudal ventral tubercle of the 6th cervical vertebra in extinct and extant *Equus*. (**a**) *Equus sp*. LACM (CIT) 138 120. (**b**) *Equus sp*. LACM (CIT) 138 120; outline of the caudal epiphyseal (red dotted line). (**c**) Partial loss of the epiphyseal in Equine Studies ‘Rideg’—3 year old male Przewalskii’s horse (Not to scale).

**Table 1 animals-13-01672-t001:** Extinct *Equus* specimens from eight museums and one educational facility.

Species (*n*); Geological Period	Institute (*n =* 8)	Collection/Catalogue No.	Adult or Subadult
*Equus sp*. (4),	NHMB	MB. Ma. 45354	Adult
		MB. Ma. 24107	Subadult
		MB. Ma. 24410	Adult
		MB. Ma. 24463	Adult
*Equus mosbachensis* (4), 80–600 kya	MMHBE	GO 374 3’ 8P 197	Adult
		Scho 13 II 4 95 690 22 14	
		Scho 13 II 4 95 690 18 45	
		Scho 13 II 4 96 692 27	
*Equus sp*. (1), 10–40 kya		Ko 100–108	Subadult
*Equus occidentalis* (20), 10–40 kya	TPMRLB	2-T22, 8’9	Adult
		E-2, 9	Subadult
		3 E2, 16	Subadult (left side only)
		3, E-1, 17.5	Adult
		3, E-2, 12	
		3, F4, 12	Subadult
		3, F-4, 12	Adult
		4 B2 M-19	
		4 B5, 20	
		4 C-2, 17–19	
		4 C-4 13–15	
		4 D2–4, 8,17	
		4 D2–99	
		4, C5, 17 246	Subadult
		60 D-12, 14–16	Adult
		4, G + H-2 + 3,8	Subadult
		67 H-9, 15–18.5	
		77 E-9, 11	Adult
		Circle 61	Subadult
		GIZ, 14.5–16.5	Adult (right side only)
*Pliohippus sp*. (1),	YHM	Mounted replica	Adult
*Hyracotherium sp*. (1),			
*Mesohippus sp*. (1),			
*Hyracotherium angustidens* (1), 55 mya	AMNH	15428	Adult (right side only)
*Merychippus isonesus* (1), 15–5 mya		ESP. 5263638	
*Mesohippus bairdii* (1), 37–32 mya		1477	Adult
*Miohippus obliquidens* (1), 32–25 mya		39115	Adult (left side only)
*Pliohippus mirabilis* (1), 12–6 mya		60810	
*Pliohippus pernix* (1), 12–6 mya		60803	Adult
*Mesohippus sp*. (cast) (1)	OUMNH	Mounted Replica	Adult
*Pliohippus sp*. (1) 12–6 mya	LACM	(CIT) 210	Adult
*Equus sp*. (6)	LACM	(CIT) 138 120	Subadult
		(CIT) 138 138	Adult (left side only)
		(CIT) 138 699	Adult
		192 L8	Adult (right side only)
	LACM San Josecito	(CIT) 192	Adult (right side only)
	LACM Row XIX	(CIT) 192 (duplicate)	Adult (right side only)
*Equus simplicidens* (15), 5.3–1.8 mya	USNM	USNM 13791	Adult
		USNM 12573	
		USNM PAL 785552	Subadult
		USNM V 12575	Adult
		USNM PAL 785561	
		USNM 785560	
		USNM PAL 785553	
		USNM 78559	
		USNM 12155	
		USNM 12580	
		USNM PAL 785557	
		USNM PAL 785556	
		USNM PAL 785559	
		USNM PAL 785555	
		USNM PAL 785554	
*Equus sp*. (7)	UF	Leisey 1A UF 82283	Adult
		Leisey 1A UF 151092	
		Leisey 1A UF 151020	Adult (right side only)
		UF 242520	
		UF 242521	Adult
		Leisey 1A UF 151094	Adult (Left side only)
		UF 242522	Adult (right side only)

**Table 2 animals-13-01672-t002:** Extant *Equus* specimens from two museums and two educational facilities.

Species (*n*); Author; Geological Period	Institute	Collection/Catalogue No.	Adult or Subadult
*Equus africanus* (1),	NBC	N.A.M. 28.-11-1972	Adult
*Equus hermionus* (1),		No. 509	
*Equus sp*. (1)		*Equus sp*. mounted skeleton	
*Equus asinus* (3),	ACEPT	Donald	Adult
		Sebastian	
		Kam	
*Equus przewalskii* (5),	ES	Heteni	
		Rideg	Subadult
	YHM	YHM—Live Pony Park	Adult
	NHMB	MB. Ma. 16464	
		MB. Ma. 45373	
*Equus quagga boehmi* (1),	ES	Kimberley—Zoo bred	

**Table 3 animals-13-01672-t003:** Extinct *Equus* specimens from three peer-reviewed publications.

Species (*n*); Author; Geological Period	Institute	Collection/Catalogue No.	Adult or Subadult
*Neohipparion whitneyi* (1), (Gidley 1903) 13.6–4.9 mya [46]	AMNH	9815	Adult
*Hyracotherium grangeri* (1), (Gingerich 1889); 55 mya [47]	UM	115547	
*Eurohippus messelensis* (1), (Haupt 1925) 56–33.9 mya [48]	SMF	SMF ME 11034	

Key: UM—University of Michigan; SMF—Senckenberg Museum und Forschungsinstitut.

**Table 4 animals-13-01672-t004:** The size of the convexity presented in the lateral profile of the ventral process in the sixth cervical process in extinct *Equus* (museum specimens and literature) in chronological order, except *Equus sp*.

Species (*n*) and Geological Period	Institute (*n* = 8)	Collection Catalogue No.	Size of Convexity
*Hyracotherium* (1)	YHM	Mounted replica	Large
*Hyracotherium* (1) 55 mya	AMNH	15428	Large
*Hyracotherium grangeri* (1) 55 mya [47]	UM	115547	Large
*Eurohippus messelensis* (1) 56–33.9 mya [48]	SMF	SMF ME 11034	*
*Mesohippus* (cast) (1)	OUMNH	Mounted Replica	Large
*Mesohippus bairdii* (1) 37–32 mya	AMNH	1477	Large
*Mesohippus* (1)	YHM	Mounted replica	Large
*Miohippus doliquideus* (1) 32–25 mya	AMNH	39115	Large
*Merychippus isonesus* (1) 15–5 mya	AMNH	ESP. 5263638	Large
*Neohipparion whitneyi* (1) 13.6–4.9 mya [46]	AMNH	9815	Large ^
*Pliohippus mirabilis* (1) 12–6 mya	AMNH	60810	Large
*Pliohippus pernix* (1) 12–6 mya	AMNH	60803	Large
*Pliohippus* (1)	YHM	Mounted replica	Large
*Pliohippus* (1) 12–6 mya	LACM	(CIT) 210	Large
*Equus simplicidens* (15) 5.3–1.8 mya	USNM	USNM 13791	Small
		USNM 12573	Small
		USNM PAL 785552	Small
		USNM V 12575	Small
		USNM PAL 785561	Small
		USNM 785560	Small
		USNM PAL 785553	Medium
		USNM 785559	Medium
		USNM 12155	Medium
		USNM 12580	Small
		USNM PAL 785557	Medium
		USNM PAL 785556	Small
		USNM PAL 785559	Small
		USNM PAL 785555	Medium
		USNM PAL 785554	Small
*Equus occidentalis* (20) 10–40 kya	TPMRLB	2-T22, 8’9	Small
		E-2, 9	Small
		3 E2, 16	Small
		3, E-1, 17.5	Medium
		3, E-2, 12	Medium
		3, F4, 12	Medium
		3, F-4, 12	Small
		4 B2 M-19	Small
		4 B5, 20	Small
		4 C-2, 17–19	Small
		4 C-4 13–15	Small
		4 D2-4, 8, 17	Small
		4 D2–99	Small
		4, C5, 17 246	Small
		60 D-12, 14–16	Small
		4, G + H-2 + 3,8	Small
		67 H-9, 15–18.5	Small
		77 E-9, 11	Medium
		Circle 61	Medium
		GIZ, 14.5–16.5	Small
*Equus mosbachensis* (4) 80–600 kya	MMHBE	GO 374 3’ 8 P 197	Small
		Scho 13 II 4 95 690 22 14	Small
		Scho 13 II 4 95 690 18 45	Small
		Scho 13 II 4 96 692 27	Small
*Equus sp*. 10–40 kya (1)		Ko 100–108	Small
*Equus sp*. (4)	NHMB	MB. Ma. 24107	Small
		MB. Ma. 24410	Medium
		MB. Ma. 24463	Medium
		MB. Ma. 45354	Medium
*Equus sp*. (6)	LACM	(CIT) 138 120	Small
		(CIT) 138 138	Small
		(CIT) 138 699	Medium
		192 L8	Medium
	LACM San Josecito	(CIT) 192	Small
	LACM Row XIX	(CIT) 192 (duplicate)	Small
*Equus sp*. (7)	UF	Leisey 1A UF 82283	Small
		Leisey 1A UF 151092	Medium
		Leisey 1A 151120	Medium
		UF 242520	Small
		UF 242521	Small
		Leisey 1A UF 151094	Medium
		UF 242522	Medium

Key: *—no convexity; ^—Gidley’s (1903) description.

**Table 5 animals-13-01672-t005:** The size of the convexity presented in the lateral profile of the ventral process in the sixth cervical process in extant *Equus* (museum specimens and educational facilities), except *E. ferus caballus*.

Species (*n*)	Institute	Collection/Catalogue No.	Adult or Subadult
*Equus africanus* (1)	NBC	N.A.M. 28.-11-1972	Small
*Equus hermionus* (1)		No. 509	
*Equus sp*. (1)		*Equus sp*. mounted skeleton	
*Equus asinus* (3)	ACEPT	Donald	Small
		Sebastian	
		Kam	
*Equus przewalskii* (5)	ES	Heteni	
		Rideg	Small
	YHM	YHM—Live Pony Park	Small
	NHMB	MB. Ma. 16464	
		MB. Ma. 45373	
*Equus quagga boehmi* (1)	ES	Kimberley	

## Data Availability

This is original research and the research data is contained within the manuscript. There is no other archived data reported elsewhere.

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
