# Peer review of "Morphology of the Ventral Process of the Sixth Cervical Vertebra in Extinct and Extant Equus: Functional Implications"

_animals, 2023, doi:10.3390/ani13101672_

Round 1

Reviewer 1 Report

Dear Authors

This is very interesting, largely original research, worthy of publication, with minor revisions. The title  fully corresponds to the content of the manuscript. As the research includes extinct taxa this topic will be of interest to the palaeontological community as well. Simple summary and abstract are informative. In keywords, enter the abbreviation next to the word Caudal ventral tubercle. If you are using a species name for the first time, introduce the name of its author. For the non-anatomists who will also be reading this paper, introduce an auxiliary drawing-location of the m.longus colli. Table 1-Note that sometimes you write Equus Sp, other times Equus sp and still others Equus sp.In Table 1, the names of species are missing from the names of their authors. Figures are sufficient and clear. Discussion and conclusions are sufficient. In the future, the experiment can be extended to include horses used in different ways in sport.

Regards

Author Response

Dear Reviewer No. 1,

Thank you for your time and expertise in reviewing this manuscript. Your comments are greatly appreciated and each has been addressed (see below).  

Responding to the more extensive comments as the minor ones were easily addressed without further ado: -

For the non-anatomists who will also be reading this paper, introduce an auxiliary drawing-location of the m.longus colli.

An addition of a Supplementary File with the relevant diagram has been placed at the end of the manuscript.    

In Table 1, the names of species are missing from the names of their authors. 

Additions in Tables 1, 2, and 3 have addressed this issue.

In the future, the experiment can be extended to include horses used in different ways in sport.

Hopefully, ongoing research will investigate this aspect.  

Once again thank you for your time and comments, they have been most helpful. 

Kind regards,

Sharon May-Davis

Reviewer 2 Report

This paper is about the equine thoracic vertebrae. the study was conducted on the modern horse and extinct species. The material used in the study is of great value. The work is important for comparative anatomy of vertebrates, but also has some practical value. The methodology of the work is typical of osteology research. The figures are of good quality and adequately complement the text.The literature can be expanded a little in relation to my opinion of the needed mention of the reasons for this and not the other structure of the transverse process in the cervical vertebrae.

[53] better use- cranial than upper

[54] better use caudal than lower

[65] It is worth mentioning that it is a costotransversarii process (proc. costotransversarius) consisting of a vestigial rib and a proper transverse process see for example in: Konig h., E., Liebich H-G.: Veterinary Anatomy of Domestic Animals. Georg Thieme Verlag, Stuttgary 2020

Author Response

Dear Reviewer No. 2,

Thank you for your time and expertise in reviewing this manuscript. Your comments are greatly appreciated and each has been addressed (see below).  

Responding to the more extensive comments as the minor ones were easily addressed without further ado: -

The literature can be expanded a little in relation to my opinion of the needed mention of the reasons for this and not the other structure of the transverse process in the cervical vertebrae.

Expanded in the 'Introduction' lines 96-98; Therefore, the aim of this study is to ascertain whether the congenital malformation of the absent CVT in C6 is a recent event in modern E. ferus caballus or a normal variation in the population by ascertaining its presence or absence in pre-domesticate Equus.   

It is worth mentioning that it is a costotransversarii process (proc. costotransversarius) consisting of a vestigial rib and a proper transverse process see for example in: Konig h., E., Liebich H-G.: Veterinary Anatomy of Domestic Animals. Georg Thieme Verlag, Stuttgary 2020

Ulitised Arnold 2021 in the 'Introduction' Lines 63-67; In C6, its atypical feature is a  pleurapophysis vestigial rib forming a bilateral ventral projection traversing the entire length of the vertebral body [1] which is referred to in the literature by several names; ventral process, ventral lamina (lamina ventralis), ventral transverse process and or ventral tubercle [6–8, 12–16, 18–29]. 

Once again thank you for your time and comments, they have been most helpful. 

Kind regards,

Sharon May-Davis

Reviewer 3 Report

This is an interesting review of evolution, anatomy, and the literature pertaining to both.

My main question is given we are evaluating a single specimen from a broad time era, what is the chance that if there were anatomic variation present we would "see" it in this limited population of extinct/ extant examples?  What is the incidence of anomalous C6 in the modern horse and even if the extinct/ extant C6 were pooled, are 83 examples enough for us to possibly find the anomaly?

I understand that this is not intended to be a review that includes those statistics specifically but it would be interesting to address it to some degree, potentially in the discussion.

Line 17: there is a period in the sentence after tubercle that in error

Line 59: extra space after vertebra

Line 142: Courtesy is misspelled

Line 242-243: I am not sure it is appropriate to mention the specimens unsuited for the study, because, as mentioned, they were not suitable for the study.  Potentially this would be better suited for the discussion.

Line 272-274: Description of the specimen with no convexity could be improved with a photo.  Given that it is a museum specimen, could a figure of this be included for the reader?

Line 319: Extra space after thus

Line 349-352: Please re-write for clarity

Excellent, see comments above for suggested corrections.

Author Response

Dear Reviewer No. 3,

Thank you for your time and expertise in reviewing this manuscript. Your comments are greatly appreciated and each has been addressed (see below).  

Responding to the more extensive comments as the minor ones were easily addressed without further ado: -

My main question is given we are evaluating a single specimen from a broad time era, what is the chance that if there were anatomic variation present we would "see" it in this limited population of extinct/ extant examples? 

At present the current extant population of Equus ferus caballus is presenting this anomalous variation at high percentages in certain breeds ('Discussion' Line 340) with percentages per breed dependant on the sample examined per study. Only 11 papers have been written on the C6 variation in modern E. ferus caballus with most reviewing retrospectively radiographic findings and these are dependant on clients presenting horses with cervical or ataxic issues. Hence, the sample is primarily skewed to compromised horses. Hopefully, the sample size in this study was sufficient to note the targeted variation in C6, yet no anomalous variations were detected in any specimen. For me these ancient Equus specimens were the most symmetrical I have ever encountered compared to extant Equus.      

What is the incidence of anomalous C6 in the modern horse and even if the extinct/ extant C6 were pooled, are 83 examples enough for us to possibly find the anomaly? I understand that this is not intended to be a review that includes those statistics specifically but it would be interesting to address it to some degree, potentially in the discussion.

See above.

Line 242-243: I am not sure it is appropriate to mention the specimens unsuited for the study, because, as mentioned, they were not suitable for the study.  Potentially this would be better suited for the discussion.

It is mentioned in the 'Discussion' and expanded minutely in the results. The explanation behind establishing this point is for the reader to understand why only 83 specimens were viewed when so many facilities had been visited. The protocols required a clear definition for observational results and this was just not an easy task when so many specimens didn't meet the criteria.     

Line 272-274: Description of the specimen with no convexity could be improved with a photo.  Given that it is a museum specimen, could a figure of this be included for the reader?

A mention of Figure 1a in the manuscript directs the reader this photo which is virtually identical to the specimen.  

Line 349-352: Please re-write for clarity

Re-worded - Through case studies based on gross examination of the thoracal Longus colli muscle, May-Davis and Walker (2015) concurred that it, either relocated with altered tendon morphology, or hypertrophied on the affected side. The authors also described potential symptomatic observations of afflicted horses bearing this anomalous variation [40]. 

I trust these responses and amendments meet your with your requirements.

Once again thank you for your time and comments, they have been most helpful. 

Kind regards,

Sharon May-Davis